# HyperPg - Prototypical Gaussians on the Hypersphere for Interpretable Deep Learning

## Abstract

Prototype Learning methods provide an interpretable alternative to black-box deep learning models. Approaches such as ProtoPNet learn, which part of a test image "look like" known prototypical parts from training images, combining predictive power with the inherent interpretability of case-based reasoning. However, existing approaches have two main drawbacks: A) They rely solely on deterministic similarity scores without statistical confidence. B) The prototypes are learned in a black-box manner without human input. This work introduces HyperPg, a new prototype representation leveraging Gaussian distributions on a hypersphere in latent space, with learnable mean and variance. HyperPg prototypes adapt to the spread of clusters in the latent space and output likelihood scores. The new architecture, HyperPgNet, leverages HyperPg to learn prototypes aligned with human concepts from pixel-level annotations. Consequently, each prototype represents a specific concept such as color, image texture, or part of the image subject. A concept extraction pipeline built on foundation models provides pixel-level annotations, significantly reducing human labeling effort. Experiments on CUB-200-2011 and Stanford Cars datasets demonstrate that HyperPgNet outperforms other prototype learning architectures while using fewer parameters and training steps. Additionally, the concept-aligned HyperPg prototypes are learned transparently, enhancing model interpretability.

## 1 Introduction

Deep Learning has achieved high accuracy in many computer vision tasks. However, the decision-making processes of these models lack transparency and interpretability, making deployment in safety-critical areas challenging. Explainable Artificial Intelligence (XAI) seeks to develop interpretability methods to open the black-box reasoning processes of these models and increase trust in their decisions.

XAI methods can be broadly divided into two categories: First, Post-Hoc Methods like LIME (Ribeiro et al., 2016), SHAP (Lundberg & Lee, 2017) or GradCAM Selvaraju et al. (2017) offer explanations for predictions without requiring retraining. While applicable in many scenarios, post-hoc methods may not actually align with the models' decision making processes, potentially leading to interpretations that are not entirely faithful (Rudin, 2019). Second, inherently interpretable methods provide built-in, case-based reasoning processes. For instance, small decision trees are inherently interpretable because their reasoning can be easily understood as a series of if-else statements (Molnar, 2020). However, they are constrained in their representational power.

Deep Prototype Learning Architectures such as ProtoPNet (Chen et al., 2019) and its derivatives (e.g., Rymarczyk et al., 2020; Donnelly et al., 2021; Sacha et al., 2023) integrate inherent interpretability into deep learning models through a prototype layer. Each neuron in this layer represents a prototype, storing a latent feature vector. The model's predictions are based on the distances between sample features and prototype parameters, for example by computing the $L_2$-distance. However, these deterministic similarity scores do not include statistical information like confidence. To include such contextual information, prototypes could be modeled as probability distributions, like Gaussian distributions. Yet, recent prototype architectures favor the cosine hypersphere as feature space for its classification advantages (Mettes et al., 2019), where defining a Gaussian distribution analogue is challenging (Hillen et al., 2017).

The prototypes in ProtoPNet and its successors are learned in an opaque manner via backpropagation. After training, the model "pushes" the learned prototypes on the embedding of the most similar known training sample (Chen et al., 2019). However, these prototypes are learned without human control and do not incorporate domain knowledge. A new training regime is needed to aligned prototypes with human-defined concepts. The main contributions of this paper are as follows:

- **HyperPg**: A new prototype representation with learned parameters anchor $\alpha$, mean $\mu$ and standard deviation $\sigma$. This representation models a Gaussian distribution over cosine similarities, thereby projecting a Gaussian distribution on the surface of a hypersphere. HyperPg's similarity score is based on the Gaussian's probability density function and adapts its size through a learned standard deviation.

- **HyperPgNet**: A new prototype learning architecture built on HyperPg prototypes. HyperPgNet learns prototypes aligned with human-defined concepts using pixel level annotations. These concepts describe features such as color and patterns of birds or car parts.

- **Concept Extraction Pipeline**: Based on Grounding DINO (Liu et al., 2023) and SAM2 (Ravi et al., 2024), this pipeline provides pixel-level concept annotations at scale.

- **Classification Experiments**: HyperPgNet is compared with other prototype learning architectures like ProtoPNet on the CUB-200-2011 (Wah et al., 2011) and Stanford Cars (Krause et al., 2013) datasets. HyperPgNet outperforms the other models with fewer learned prototypes, while infusing each prototype with more meaning.

## 2 RELATED WORK

**Prototype Learning.** In image classification, prototype learning approaches using autoencoders provide high interpretability by reconstructing learned prototypes from latent space back to the image space (Li et al., 2018). However, these approaches are limited in their performance because each prototype must represent the entire image. ProtoPNet (Chen et al., 2019) introduced the idea of prototypical parts. In this setting, each prototype is a latent patch of the input image, commonly a $1 \times 1$ latent patch. The prototypes are each associated with a single class and learned via backpropagation without additional information.

HyperPgNet builds on the idea of prototypical parts, but learns the prototypes in a transparent, concept-aligned manner. The prototypes are not class exclusive, but class shared. Each prototype corresponds to a human defined concept. To provide the required concept annotations, we propose a labeling pipeline based on foundation models such as DINOv2 (Oquab et al., 2024) and SAM2 (Ravi et al., 2024). HyperPgNet learns the concept-aligned prototypes by only adapting the loss functions and providing additional annotations, without requiring changes to the feature encoder like PIPNet (Nauta & Seifert, 2023) or Lucid-PPN (Pach et al., 2024), which uses a hybrid input head to disentangle prototype color and shape.

Multiple successors build on the idea of ProtoPNet. ProtoPShare (Rymarczyk et al., 2020) and ProtoPool (Rymarczyk et al., 2022) use additional optimizations to learn class-shared prototypical parts. Deformable ProtoPNet (Donnelly et al., 2021) learns a mixture of prototypical parts with dynamic spatial arrangement. Other work replace the linear output layer of ProtoPNet with other interpretable models: ProtoKNN (Ukai et al., 2023) uses a k-nearest neighbor classifier and ProtoTree (Nauta et al., 2021) employs a decision tree.

All these models change how the prototype activations are further processed. However, they all rely on the point-based prototype formulation introduced by ProtoPNet based on the $L_2$ or cosine similarity. HyperPgNet proposes a novel prototype formulation based on Gaussian distributions on the hypersphere, without changing the downstream processing. This probabilistic view with learned mean and variance allows HyperPgNet to learn prototypes with different degrees of specialization. Furthermore, the statistical confidence derived from the learned variance could be integrate in further downstream processing.

The view of prototype learning as clustering in latent space was introduced for image segmentation (Zhou et al., 2022). The image patches are points in latent space with the prototypes as cluster centers. ProtoGMM (Moradinasab et al., 2024) and MGProto (Wang et al., 2023) build on this idea to model the prototypes as Gaussian mixture models for image segmentation.

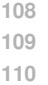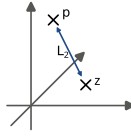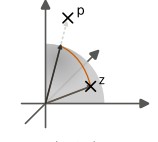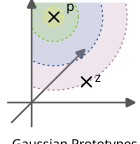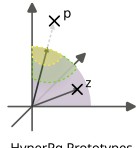

Figure 1: Illustration on how the different prototype formulations compute the similarity between a prototype $p$ and latent vector $z$. $L_2$ prototypes compute the Euclidean distance between two points in latent space. Hyperspherical prototypes use the cosine similarity of normalized vectors, which corresponds to the angle between two points on a hypersphere. Gaussian prototypes model a Gaussian distribution in Euclidean space and compute the probability density function (PDF). HyperPg prototypes learn a Gaussian distribution of cosine similarities, thereby projecting a Gaussian distribution onto the surface of a hypersphere.

HyperPg adapts the idea of Gaussian prototypes used in image segmentation to hyperspherical latent spaces based on the cosine similarity. The cosine similarity has been shown to perform well in classification tasks (Mettes et al., 2019) and also been applied in recent prototype learning works (Zhou et al., 2022; Ukai et al., 2023). This makes HyperPgNet the first model to employ probabilistic prototypes with learnable mean and variance in a hyperspherical space.

**Concept-Based Methods.** Testing with Concept Activation Vectors (TCAV) (Kim et al., 2018) is a post-hoc method that aligns model activations with human-understandable concepts. In image classification, TCAV measures model activations on images representing a specific concept like "striped" and compares them to activations on a random image set. Concept Activation Regions (CAR) (Crabbé & van der Schaar, 2022) relax TCAV's assumption of linear separability in latent space by using support vector machines and the kernel trick. Another post-hoc approach, CRAFT (Fel et al., 2023), generates concept attribution maps by identifying relevant concepts from the training set via Non-Negative Matrix Factorization (NMF) and backpropagating with GradCAM (Selvaraju et al., 2017).

Instead of analyzing the learned concepts post-hoc, some approaches integrate concept learning into the classifier. MCPNet (Wang et al., 2024) extracts concepts from multiple layers by splitting feature maps along the channel dimension, learning concepts during training and at multiple scales. Concept Bottleneck Models (CBMs) (Koh et al., 2020) augment black-box models to predict pre-defined concept labels in an intermediate step. In this manner, CBMs behave like two sequential black box models and their internal reasoning remains opaque.

HyperPgNet combines the concept-aligned learning approach of CBM with interpretable prototype learning. HyperPgNet learns multiple HyperPg prototypes per concept, instead of a fully connected concept prediction layer. A novel concept extraction pipeline, built on foundation models, provides pixel level annotations at scale. By using those annotations to learn concept prototypes, HyperPgNet avoids the ambiguity of aligning post-hoc explanations with the model's internal reasoning.

## 3 PROTOTYPICAL GAUSSIANS ON THE HYPERSPHERE

Prototype Learning is an inherently interpretable machine learning method. The reasoning process is based on the similarity scores of the inputs to the prototypes, retained representations of the training data. For example, a K-Nearest Neighbor (KNN) model is a prototype learning approach with the identity function for representation and an unlimited number of prototypes. In contrast, a Gaussian Mixture Model (GMM) uses a mean representation but restricts the number of prototypes to the number of mixture components.

Prototype learning for deep neural networks involves finding structures in latent space representations. This section provides an overview of existing methods, which are also illustrated in Fig. 1. Prior work uses point-based prototypes, computing similarity scores relative to a single point in latent space. Aligning prototypes with human-labeled concepts requires a more powerful prototype representation, leading to the introduction of HyperPg - Prototypical Gaussians on the Hypersphere.

HyperPg prototypes are able to adapt their shape in the latent space by modeling Gaussian distributions with mean and standard deviation, adapting to the variance in encoded concept features.

## 3.1 POINT BASED PROTOTYPES

The general formulation of prototypes, as defined in previous work (e.g., Chen et al., 2019), is discussed first. Let $\mathcal{D} = [\boldsymbol{X}, \boldsymbol{Y}] = \{(\boldsymbol{x}_i, y_i)\}_{i=1}^N$ denote the training set, e.g., a set of labeled images, with classes $C$. Each class $c \in C$ is represented by $Q$ many prototypes $\boldsymbol{P}_c = \{\boldsymbol{p}_{c,j}\}_{j=1}^Q$.

Some feature encoder Enc projects the inputs into a $D$-dimensional latent space $\mathcal{Z}$, with $\boldsymbol{z}_i = \text{Enc}(\boldsymbol{x}_i)$ being a feature map of shape $\zeta_w \times \zeta_h \times D$ with spatial size $\zeta = \zeta_w \zeta_h$. Commonly, the prototypes $\boldsymbol{p}$ are also part of $\mathcal{Z}$ with shape $\rho_w \times \rho_h \times D$, i.e., spatial size $\rho = \rho_w \rho_h$.

Autoencoder approaches use $\rho = \zeta$, meaning the prototype represents the entire image and can be reconstructed from latent space (Li et al., 2018). Part-based approaches like ProtoPNet and segmentation models like ProtoSeg use $\rho = 1$ (Chen et al., 2019; Zhou et al., 2022), meaning each prototype represents some part of the image. Notable exceptions include Deformable ProtoPNet (Donnelly et al., 2021), where each prototype has spatial size $\rho = 3 \times 3$ and MCPNet (Wang et al., 2024), where prototypes are obtained by dividing the channels of the latent space into chunks.

The prediction is computed by comparing each prototype $\boldsymbol{p}$ to the latent feature map $\boldsymbol{z}$. For simplicity's sake lets assume the spatial dimensions $\rho = \zeta = 1$. The following equations can be adapted for higher spatial dimensions by summing over $\sum_{\rho_w} \sum_{\rho_h}$ for each chunk of the latent map.

ProtoPNet's prototypes leverage the $L_2$ similarity. The $L_2$ similarity measure is defined as

$$s_{L_2}(\boldsymbol{z}|\boldsymbol{p}) = \log\left(\frac{\|\boldsymbol{z} - \boldsymbol{p}\|_2^2 + 1}{\|\boldsymbol{z} - \boldsymbol{p}\|_2^2 + \epsilon}\right) \tag{1}$$

and is based on the inverted $L_2$ distance between a latent vector $\boldsymbol{z}$ and a prototype vector $\boldsymbol{p}$. This similarity is a point-based measure, as only the two vectors are compared, without any additional context like the expected variance of the cluster represented by the prototype.

Hyperspherical prototypes using the cosine similarity have been shown to perform well in classification tasks (Mettes et al., 2019) and have been widely used since (e.g., Zhou et al., 2022; Ukai et al., 2023). The cosine similarity is defined as

$$s_{\cos}(\boldsymbol{z}|\boldsymbol{p}) = \frac{\boldsymbol{z}^\top \boldsymbol{p}}{\|\boldsymbol{z}\|_2 \|\boldsymbol{p}\|_2}, \tag{2}$$

which is based on the angle between two normalized vectors of unit length. By normalizing $D$ dimensional vectors to unit length, they are projected onto the surface of a $D$ dimensional hypershere. The cosine similarity is defined on the interval $[-1, 1]$ and measures: 1 for two vectors pointing in the same direction, 0 for orthogonal vectors, and $-1$ for vectors pointing in opposite directions. Like the $L_2$ similarity, the cosine similarity is a point-based measure comparing only two vectors.

Both the $L_2$ and cosine similarity can be used for classification. The similarity scores are processed by a fully connected layer (e.g., Chen et al., 2019; Donnelly et al., 2021), or a winner-takes-all approach assigns the class of the most similar prototype (e.g., Sacha et al., 2023). Prototypes can be learned by optimizing a task-specific loss, such as cross-entropy, via backpropagation. Alternatively, Zhou et al. (2022) propose "non-learnable" prototypes, whose parameters are obtained via clustering in the latent space rather than backpropagation.

## 3.2 GAUSSIAN PROTOTYPES

Gaussian prototypes model prototypes as a Gaussian distribution with mean and covariance. They adapt to the spread of the associated latent cluster by adjusting their covariance matrix. Thus, a Gaussian prototype with a wide covariance can still have a relatively high response even for larger distances from the mean vector.

Let the formal definition of a Gaussian prototype be $\boldsymbol{p}^G = (\boldsymbol{\mu}, \boldsymbol{\Sigma})$. The parameters of $\boldsymbol{p}_{c,j}^G$ now track both the mean and covariance of latent vector distribution $\boldsymbol{Z}_{c,j}$. Each Gaussian prototype $\boldsymbol{p}^G$ thus defines a multivariate Gaussian Distribution $\mathcal{N}(\boldsymbol{\mu}, \boldsymbol{\Sigma})$. Gaussian prototypes can be trained using

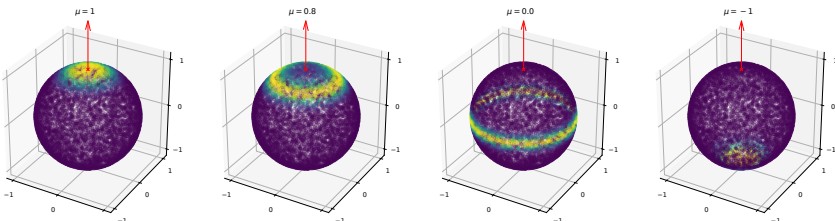

Figure 2: HyperPg learns a Gaussian distribution with mean $\mu$ and std $\sigma$ of cosine similarities with the anchor $\boldsymbol{\alpha}$. The plots visualize HyperPg's activations on the $3D$ hypersphere with anchor $\boldsymbol{\alpha} = (0, 0, 1)$, std $\sigma = 0.1$ and and various mean cosine similarities $\mu \in [-1, 1]$ for 10k random samples. The anchor $\boldsymbol{\alpha}$ is shown as a red arrow, and std $\sigma$ governs the width of the distribution. For $\mu = 1$, the distribution is aligned with the vector; for $\mu = -1$ it is on the opposite side of the hypersphere. Due to the cosine similarity, setting $1 > \mu > -1$ interpolates the area of highest probability density between both poles, leading to a ring shape on the hypersphere's surface.

EM for clustering in the latent space (Zhou et al., 2022; Wang et al., 2023; Moradinasab et al., 2024) or by directly optimizing the parameters via Backpropagation. The similarity measure of Gaussian prototypes is defined as the probability density function (PDF) for $D$-dimensional multivariate Gaussians, namely

$$s_{\text{Gauss}}(\boldsymbol{z}|\boldsymbol{p}^G) = \mathcal{N}(\boldsymbol{z}; \boldsymbol{\mu}, \boldsymbol{\sigma}) \tag{3}$$

This formulation as a PDF has several advantages: A) The similarity can be interpreted as the likelihood of being sampled from the Gaussian prototypes, which is more meaningful than a distance metric in a high-dimensional latent space. B) Prototypes can adapt their shape using a full covariance matrix, allowing different variances along various feature dimensions, offering more flexibility in shaping the latent space. However, this increases computational requirements, especially with EM clustering. Gaussians prototypes lose the advantages of hyperspherical prototypes for classification and regression Mettes et al. (2019).

### 3.3 GAUSSIAN PROTOTYPES ON THE HYPERSPHERE - HYPERPG

Prototypical Gaussians on the Hypersphere (HyperPg) combine the advantages of Gaussian and hyperspherical prototypes. HyperPg prototypes are defined as $\boldsymbol{p}^H = (\boldsymbol{\alpha}, \mu, \sigma)$ with a directional anchor vector $\boldsymbol{\alpha}$, *scalar* mean similarity $\mu$ and *scalar* standard deviation (std) $\sigma$. HyperPg prototypes learn a $1D$ Gaussian distribution over the cosine similarities to the anchor vector $\boldsymbol{\alpha}$. Because the cosine similarity is bounded to $[-1, 1]$, HyperPg's similarity measure is defined as the PDF of the truncated Gaussian distribution within these bounds. Let $\mathcal{G}(x, \mu, \sigma)$ be the cumulative Gaussian distribution function. Then, HyperPg's similarity measure based on the truncated Gaussian distribution is defined as

$$s_{\text{HyperPg}}(\boldsymbol{z}|\boldsymbol{p}^H) = \mathcal{T}_G(s_{\cos}(\boldsymbol{z}|\boldsymbol{\alpha}); \mu, \sigma, -1, 1) \tag{4}$$

$$= \frac{\mathcal{N}(s_{\cos}(\boldsymbol{z}|\boldsymbol{\alpha}); \mu, \sigma)}{\mathcal{G}(1, \mu, \sigma) - \mathcal{G}(-1, \mu, \sigma).} \tag{5}$$

Fig. 2 illustrates the activations of HyperPg's similarity function on the surface of a $3D$ hypersphere with anchor $\boldsymbol{\alpha} = (0, 0, 1)$, fixed std $\sigma = 0.1$ and various mean values $\mu \in [-1, 1]$. The anchor $\boldsymbol{\alpha}$ defines a prototypical direction vector in latent space $\mathcal{Z}$, similar to other hyperspherical prototypes, and is visualized as a red arrow. The learned Gaussian distribution of cosine similarities is projected onto the hypersphere's surface with std $\sigma$ governing the spread of the distribution, and mean $\mu$ the expected distance to the anchor $\boldsymbol{\alpha}$. For $\mu = 1$, the distribution centers around the anchor, as the cosine similarity is 1 if two vectors point in the same direction. For $\mu = -1$, the distribution is the on opposite side of the hypersphere, as the cosine similarity is $-1$ for vectors pointing in opposite directions.

For values of $1 > \mu > -1$, the distribution forms a hollow ring around the anchor vector $\boldsymbol{\alpha}$. This occurs because the cosine similarity for these $\mu$ values expects the activating vectors to point in a

different direction than the anchor, without specifying the direction. Imagining the interpolation between $\mu = 1$ and $\mu = -1$, the probability mass moves from one pole of the hypersphere to the other, stretching like a rubber band over the surface. For $\mu = 0$, the expected cosine similarity indicates that vectors with the highest activation are orthogonal to the anchor $\boldsymbol{\alpha}$. Since no specific direction is indicated, the entire hyperplanar segment orthogonal to the anchor has the highest activation. This activation pattern for the cosine similarity would typically require an infinite mixture of prototype vectors pointing in all directions in this hyperplane. HyperPg achieves the same effect by learning only one prototype vector (the anchor) and just two additional scalar parameters. This significantly increases HyperPg's representational power compared to standard hyperspherical and Gaussian prototypes and is a major difference to the von Mises-Fisher distribution (see also Appendix A).

Combining the strengths of hyperspherical and Gaussian prototypes, HyperPg can learn more complex structures in the latent space, such as human-defined concepts, and provide a more meaningful similarity measure. HyperPg can be easily adapted to other probability distributions with additional desirable properties. Possible candidate distributions are elaborated on in Appendix A. Similarly, it is possible to exchange the cosine similarity to other similarity measures or functions, and learn an untruncated PDF over their output, making the HyperPg idea transferable to applications outside of prototype learning.

## 4 TRAINING

The original ProtoPNet implementation uses three loss functions: a task specific loss like crossentropy for classification, a cluster loss to increase compactness within a class's cluster, and a separation loss to increase distances between different prototype clusters. However, the prototypes of ProtoPNet and its successors are learned in a black-box manner.

HyperPgNet is a new inherently interpretable deep learning approach built on HyperPg prototypes. It introduces a "Right for the Right Concept" loss, inspired by "Right for the Right Reasons" (Ross et al., 2017), to restrict the learned prototypes to human-defined concepts. This focus enhances the interpretability and minimizes the influence of confounding factors. This section first provides an overview of the used prototype learning losses, then introduces the Right for the Right Concept loss, and finally discusses the overall network architecture and final multi-objective loss.

### 4.1 PROTOTYPE LOSSES

ProtoPNet defines a cluster loss function to shape the latent space such that all latent vectors $\boldsymbol{z}_c \in \boldsymbol{Z}_c$ with class label $c$ are clustered tightly around the semantically similar prototypes $\boldsymbol{p}_c \in \boldsymbol{P}_c$. The cluster loss function is defined as

$$L_{\text{Clst}} = -\frac{1}{N} \sum_{i=1}^{N} \frac{1}{|C|} \sum_{c \in C} \max_{\boldsymbol{p}_c \in \boldsymbol{P}_c} \max_{\boldsymbol{z}_{c,i} \in \boldsymbol{Z}_{c,i}} s(\boldsymbol{p}_c, \boldsymbol{z}_{c,i}), \quad (6)$$

where $s(\cdot, \cdot)$ is some similarity measure. The $L_{\text{Clst}}$-Loss function increases compactness by increasing the similarity between prototypes $\boldsymbol{p}_c$ of class $c$ latent embeddings $\boldsymbol{z}_c$ of class $c$ over all samples.

An additional separation loss increases the margin between different prototypes. The separation loss function is defined as

$$L_{\text{Sep}} = \frac{1}{N} \sum_{i=1}^{N} \frac{1}{|C|} \sum_{c \in C} \max_{\boldsymbol{p}_{\neg c} \notin \boldsymbol{P}_c} \max_{\boldsymbol{z}_{c,i} \in \boldsymbol{Z}_{c,i}} s(\boldsymbol{p}_{\neg c}, \boldsymbol{z}_{c,i}), \quad (7)$$

The $L_{\text{Sep}}$ function punishes high similarity values between a latent vector $\boldsymbol{z}_c$ of class c and prototypes $\boldsymbol{p}_{\neg c}$ not belonging to $c$, thereby separating the clusters in latent space. Please note, ProtoPNet (Chen et al., 2019) use a slightly different notation by working with the $L_2$ *distance*, instead of a *similarity* measure.

In HyperPgNet the learned prototypes are not class exclusive, but shared among different classes. Instead, each prototype is assigned to one human-defined concept $k \in K$. The Concept Activation

Figure 3: HyperPgNet Architecture. The HyperPg module can be easily exchanged to other prototype formulations such as ProtoPNet. HyperPgNet uses the truncated Gaussian distribution as density estimator, but other PDFs are possible.

Regions (CAR, Crabbé & van der Schaar, 2022) model proposes a concept density loss defined as

$$L_{Density} = -\frac{1}{|K|} \sum_{k \in K} \frac{1}{|\boldsymbol{P}_k|} \sum_{\boldsymbol{p}_k \in \boldsymbol{P}_k} \phi(\boldsymbol{p}_k, \boldsymbol{Z}_k) - \phi(\boldsymbol{p}_k, \boldsymbol{Z}_{\neg k}) \tag{8}$$

$$\phi(\boldsymbol{p}, \boldsymbol{Z}) = \frac{1}{|\boldsymbol{Z}|} \sum_{\boldsymbol{z} \in \boldsymbol{Z}} s(\boldsymbol{p}, \boldsymbol{z}). \tag{9}$$

The density loss uses the similarity aggregation $\phi(\boldsymbol{p}, \boldsymbol{Z})$ which computes the mean response of a prototype $\boldsymbol{p}$ with a set of latent features $\boldsymbol{Z}$. The density loss therefore computes the mean response of correctly assigned prototypes minus the mean response of incorrectly assigned prototypes. The loss functions proposed by ProtoPNet compute the *maximum* response of correctly and incorrectly assigned prototypes instead.

## 4.2 RIGHT CONCEPT LOSS

To ensure prototypes actually correspond to the input pixels containing the concept, and do not respond to other factors in the background, HyperPgNet introduces the "Right for the Right Concept" (RRC) -Loss inspired by "Right for the Right Reasons" (RRR, Ross et al., 2017). The RRC loss is defined as

$$L_{\text{RRC}} = \frac{1}{N} \sum_{i=1}^{N} \sum_{k \in K} \left( \boldsymbol{A}_{\boldsymbol{x}_i,k} \frac{\partial}{\partial \boldsymbol{x}_i} \sum_{\boldsymbol{p}_k \in \boldsymbol{P}_k} s\left(\boldsymbol{p}_k, \text{Enc}\left(\boldsymbol{x}_i\right)\right) \right)^2, \tag{10}$$

with binary annotation matrix $\boldsymbol{A}_{\boldsymbol{x}_i,k} \in \{0,1\}^{N \times \times W \times H}$ for each input sample $\boldsymbol{x}_i$ and concept $k$. This annotation matrix defines for each input image, which pixels contain which concept. In the original RRR paper, the annotation matrix specified relevant regions for the classification task, steering the model's activations away from confounding factors in the background. In HyperPgNet the RRC-Loss further strengthens the prototype-concept association.

## 4.3 MULTI-OBJECTIVE LOSS FUNCTION

To train a prototype learning network like HyperPgNet for downstream tasks like image classification, a multi-objective loss function is employed. This multi-objective loss function is defined as

$$L = L_{\text{CE}} + \lambda_{\text{Clst}} L_{\text{Clst}} + \lambda_{\text{Sep}} L_{\text{Sep}} + \lambda_{\text{RRC}} L_{\text{RRC}},$$

where $L_{\text{CE}}$ is the cross-entropy loss over network predictions and ground truth image labels. ProtoPNet uses $\lambda_{\text{Clst}} = 0.8$ and $\lambda_{\text{Sep}} = 0.08$ (Chen et al., 2019). For HyperPgNet, the different loss terms are weighted equally, meaning all $\lambda = 1$.

## 4.4 NETWORK ARCHITECTURE

Fig. 3 illustrates HyperPgNet's Architecture for interpretable image classification. HyperPgNet uses a pretrained feature encoder such as ConvNext-tiny (Liu et al., 2022) or MiT-B4 (Xie et al., 2021) as a backbone model. A neck consisting of two $1 \times 1$ convolution layers with ReLU activation in between projects the high dimensional feature map of the backbone into a lower dimensional feature space.

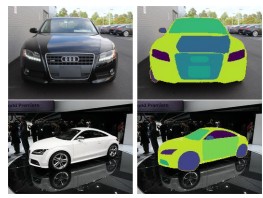

(a) Concept Segmentation Masks for CUB-200-2011 based on provided positive and negative part annotations.

(b) Concept Segmentation Mask for Stanford Cars generated from a list of human-defined car parts.

Figure 4: Examples for Automated Concept Extraction.

The HyperPg prototype module is implemented with two layers: First, a prototype learning layer computes the cosine similarity of the learnable HyperPg anchors $\alpha$ to the latent vectors produced by the neck. Second, a Density Estimation layer with learnable parameters mean $\mu$ and std $\sigma$ computes the Gaussian PDF over the activations of the previous layer. Future extensions can adapt both components independently, for example by implementing a hyperbolic similarity measure or a multi-modal probability distribution.

Finally, the output of the HyperPg Module is passed through a single fully connected layer (FCL) to produce the output logits or class scores.

## 5 CONCEPT EXTRACTION AT SCALE

To learn prototypes based on human-defined concepts with RRC-Loss for image classification, pixel level annotations are required. Foundation models are leveraged to generate these annotations, reducing the need for human labeling.

The CUB-200-2011 (Wah et al., 2011) annotates each image with up to 15 part locations (e.g., head, tail, wing). Each part is labeled with some attribute describing the coloring, pattern or shape. A Segment Anything 2 (SAM2) model (Ravi et al., 2024) creates pixel-level segmentation masks based on the part locations.

The Stanford Cars dataset (Krause et al., 2013) does not provide part level annotations. We defined a list of 10 car parts (e.g., wheel, headlight, radiator) and use a Grounding DINO model (Liu et al., 2023) to generate bounding boxes for each part. Finally, SAM2 creates pixel-level segmentation masks based on the bounding boxes. This automated pipeline labeled the 16k images of Stanford Cars within 2h on a NVIDIA 4060Ti with 16GB VRAM, and can be easily adapted to other datasets. Fig. 4 presents some examples for CUB-200-2011 and Stanford Cars.

## 6 EXPERIMENTS

The experiments were performed on two datasets: CUB-200-2011 (CUB) with 200 bird species Wah et al. (2011) and Stanford Cars (CARS) with 196 car models Krause et al. (2013). Implementation details such as data preprocessing or model implementation are provided in Appendix B.

|  |  | **Birds** | | | **Cars** | | |
|---|---|---|---|---|---|---|---|
|  |  | **# Prototypes** | **MiT-B4** | **ConvNeXt** | **# Prototypes** | **MiT-B4** | **ConvNeXt** |
| BB | Baseline | - | 17.7 | 74.2 | - | 1.9 | 57.5 |
|  | CBM | - | 75.7 | **77.9** | - | 79.9 | 81.0 |
| PP | ProtoPNet (PPN) | 2000 | 68.0 | 68.1 | 1960 | 86.4 | 87.0 |
|  | PPN + HyperPG | 2000 | 70.5 | 65.0 | 1960 | 87.4 | 77.3 |
| CAP | HyperPgNet $-L_{\text{RRC}}$ | 300 | **76.5** | 76.9 | 180 | **88.6** | **88.9** |
| (Ours) | HyperPgNet $+L_{\text{RRC}}$ | 300 | 74.1 | 71.5 | 180 | 81.2 | 86.0 |

Table 1: Test Top-1 Accuracy. Models are grouped by their reasoning process. BB: Black Box. PP: Prototypical Parts. CAP: Concept Aligned Prototypes.

## 6.1 QUANTITATIVE RESULTS

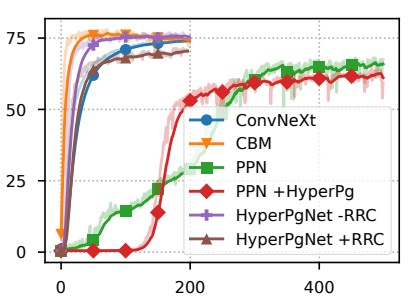

Figure 5: CUB Test Accuracy per Epoch.

Table 1 reports the top-1 accuracy for the tested models on the CUB and CARS datasets with different backbones. HyperPgNet achieves comparable results with the black box CBM and baseline models. Notably, HyperPgNet outperforms CBM on CARS, as it can differentiate between different modes for the same concept by learning multiple prototypes per concept. CBM is restricted to a binary concept presence prediction. Including $\mathcal{L}_{\mathrm{RRC}}$ slightly lowers HyperPgNet's performance but is still comparable to the black-box models.

Fig. 5 shows the test accuracy over epochs for CUB with ConvNeXt backbone. The concept aligned models CBM and HyperPgNet are the fastest to converge at around 100 epochs. ProtoPNet requires over 400 training epochs until the best performance level is reached. However, switching from the standard $L_2$ based prototypes to HyperPg prototypes effectively halves the required training epochs.

## 6.2 QUALITATIVE COMPARISON

Fig. 7 shows the pixel attribution of the highest activated prototype on each image for the tested prototype learning models with ConvNeXt backbone. In contrast to prior work (e.g., Chen et al., 2019; Ukai et al., 2023), this visualization is based on the prototype gradients, similar to saliency maps such as GradCAM (Selvaraju et al., 2017), which should avoid spatial misalignment often observed in prototype explanations (Sacha et al., 2024). The concept aligned prototypes learned by HyperPgNet focus more on the relevant parts of the image subject than the class-based prototypes of ProtoPNet. The network seems to focus correctly on either the overall color of the bird and ignoring the background (first three examples) or on specific markings (last three examples). The addition of the $\mathcal{L}_{\mathrm{RRC}}$ loss seems to further increase the main focus of the prototype activation on the relevant image regions. For ProtoPNet, no qualitative difference can be detected between the standard $L_2$ or HyperPg prototypes. With both prototype formulations, ProtoPNet is likely to also activate on parts of the image background. This is a fundamental weakness of training ProtoPNet in an opaque manner without alignment to relevant concepts, which HyperPgNet addresses.

On the CARS dataset, the effect of $\mathcal{L}_{\mathrm{RRC}}$ is more apparent. The defined concepts for this dataset are limited to car parts without fine-grained differentiation between patterns or shapes, such as the concepts on CUB. For this dataset, one of the most predictive image regions across models seems to be the front bumper of the cars, which has not been annotated with its own concept. As shown in Fig. 6, the concept-aligned prototypes still activate in this region. However, by training the model with the $\mathcal{L}_{\mathrm{RRC}}$ loss, the prototypes focus more on the annotated image regions.

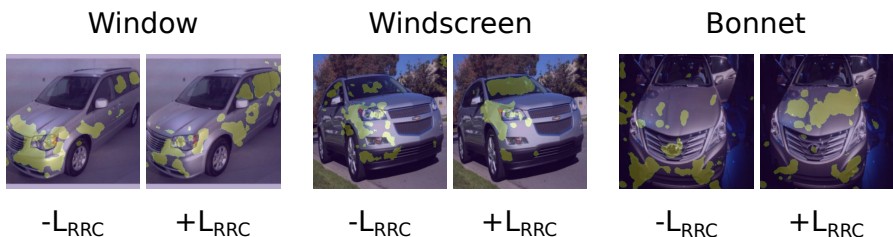

Window      Windscreen      Bonnet

$-L_{\mathrm{RRC}}$   $+L_{\mathrm{RRC}}$    $-L_{\mathrm{RRC}}$   $+L_{\mathrm{RRC}}$    $-L_{\mathrm{RRC}}$   $+L_{\mathrm{RRC}}$

Figure 6: Concept prototype activation after training with and without $\mathcal{L}_{\mathrm{RRC}}$ with MiT-B4 backbone.

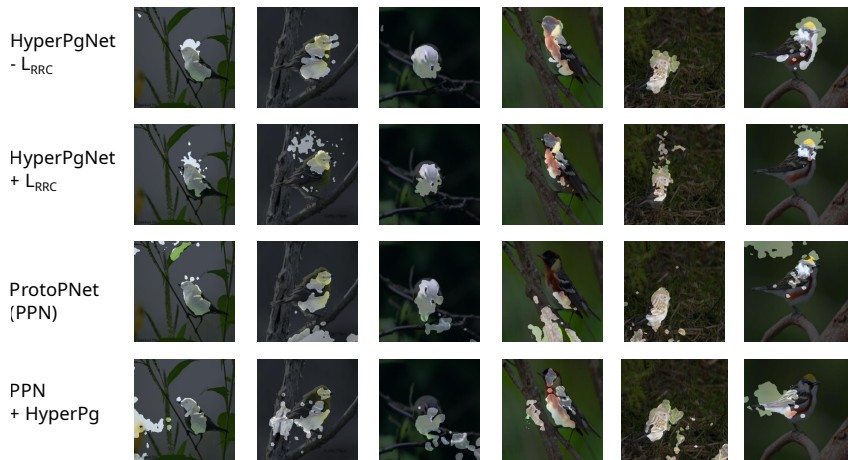

Figure 7: Highest prototype activation per image with ConvNeXt backbone.

### 6.3 ABLATION

We conduct an ablation on the number of prototypes per concept. Table 2 presents the test accuracies on CUB for HyperPgNet with ResNet50 backbone. The difference in task performance over the different model configurations is negligible. However, looking at some qualitative examples as depicted in Fig. 8 a difference in prototype quality can be observed. With fewer prototypes per concept, the model is more likely to also react to parts of the mage background in order to still achieve a low training error. With too many prototypes per concept, the latent space becomes too segmented. The prototypes start to focus on smaller subparts of the image subject, potentially decreasing the interpretability. The chosen configuration of 10 prototypes per concept seems to be a sweet spot regarding predictive performance and prototype quality for the experiments.

| # Prototypes | CUB Accuracy |
|---|---|
| CBM | 71.6 |
| 1 | 70.5 |
| 5 | 68.9 |
| 10 | 68.0 |
| 20 | 67.3 |

Table 2: Test Top-1 Accuracy on CUB with ResNet50 backbone with varying numbers of prototypes per concept.

| 1 Prototype | 5 Prototypes | 10 Prototypes | 20 Prototypes |
|---|---|---|---|

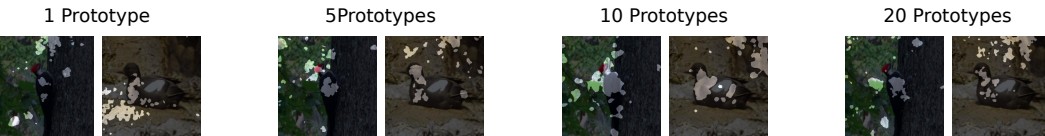

Figure 8: Qualitative Results for HyperPgNet with ResNet50 backbone and varying numbers of prototypes per concept.

## 7 CONCLUSION

This work introduces HyperPg, a new prototype representation learning a probability distribution on the surface of a hypersphere in latent space. HyperPg prototypes adapt to the variance of clusters in latent space and improve training time and accuracy compared to other prototype formulations. HyperPgNet leverages HyperPg to learn human-defined concept prototypes instead of black-box optimized prototypes. The combination of probabilistic prototypes on the hypersphere and concept-aligned prototypes allows HyperPgNet to outperform other prototype learning approaches with regards to accuracy and interpretability. One limitation HyperPgNet faces are slightly higher computational requirements due to the inclusion of concept annotations and $L_{RRC}$ during training. However, this is offset by faster convergence. Coupled with the increased transparency and interpretability of the model, this makes HyperPgNet a strong contender for scenarios with higher requirements for model trust and safety, like medical applications or human-robot interaction.

REPRODUCIBILITY STATEMENT

For reproducing the experiments, please refer to the following sections: Sec. 4 describes the required loss functions and the overall network architecture. Subsection 3.3 details the activation function of the newly introduced HyperPg prototypes. Appendix B provides detailed information on the experiment implementations, including training hyperparameters, processing of the datasets and model implementation details. Furthermore, we will publish the code on GitHub after the double-blind review period.

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

## A    ADAPTING HYPERPG TO OTHER PROBABILITY DISTRIBUTIONS

Subsection 3.3 defines HyperPg prototypes $\boldsymbol{p}^H = (\boldsymbol{\alpha}, \mu, \sigma)$ as a Gaussian Distribution with mean $\mu$ and std $\sigma$ of cosine similarities around an anchor vector $\alpha$. This idea of learning a distribution of cosine similarity values around an anchor $\boldsymbol{\alpha}$ can be adapted to other distributions. This sections introduces some potential candidates. As early experiments on the CUB-200-2011 dataset showed no significant difference in performance, these sections are relegated to the appendix.

### A.1    CAUCHY DISTRIBUTION

One theoretical disadvantage of the Gaussian distribution is the fast approach to zero, which is why a distribution with heavier tails such as the Cauchy distribution might be desirable. The Cauchy distribution's PDF is defined as

$$\mathcal{C}(x; x_0, \gamma) = \frac{1}{\pi\gamma\left(1 + \left(\frac{x-x_0}{\gamma}\right)^2\right)}, \tag{11}$$

with median $x_0$ and average absolute deviation $\gamma$. The HyperPg prototypes with Cauchy are defined as accordingly as $\boldsymbol{p}^{\text{Cauchy}} = (\boldsymbol{\alpha}, x_0, \gamma)$.

Fig. 9 illustrates the PDF of the Gaussian and Cauchy distributions with $\mu = x_0 = 1$ and $\sigma = \gamma = 0.2$, i.e., the main probability mass is aligned with the anchor $\boldsymbol{\alpha}$. The Gaussian distributions PDF quickly approaches zero and stays near constant. This could potentially cause vanishing gradient issues during training. The heavier tails of the Cauchy distribution ensure that for virtually the entire value range of the cosine similarity, gradients could be propagated back through the model. However, experiments on CUB-200-2011 showed no significant performance difference between using HyperPg with the Gaussian or Cauchy distribution.

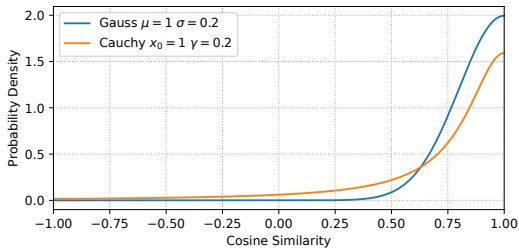

Figure 9: PDF for the Gaussian and Cauchy distribution of cosine similarity values. The Cauchy distribution has heavier tails, avoiding vanishing gradients issues.

### A.2    TRUNCATED DISTRIBUTIONS

The cosine similarity is defined only on the interval $[-1, 1]$. This makes it attractive to also use truncated probability distributions, which are also only defined on this interval. The truncation imposes a limit on the range of the PDF, thereby limiting the influence of large values for the distribution's $\sigma$ or $\gamma$ parameter, respectively. The truncated Gaussian pdf $\mathcal{T}_{\text{Gauss}}$ requires the cumulative probability function $\mathcal{G}$ and error function $f_{\text{err}}$, and is defined as

$$f_{\text{err}}(x) = \frac{2}{\sqrt{\pi}} \int_0^x \exp\left(-z^2\right) dz, \tag{12}$$

$$\mathcal{N}(x; \mu, \sigma) = \frac{1}{\sqrt{2\pi\sigma^2}} \exp\left(-\frac{(x-\mu)^2}{2\sigma^2}\right), \tag{13}$$

$$\mathcal{G}(x, \mu, \sigma) = \frac{1}{2}\left(1 + f_{\text{err}}\left(\frac{x-\mu}{\sigma\sqrt{2}}\right)\right), \tag{14}$$

$$\mathcal{T}_{\text{Gauss}}(x, \mu, \sigma, a, b) = \frac{\mathcal{N}\left(s_{\cos}(\boldsymbol{z}|\boldsymbol{\alpha}); \mu, \sigma\right)}{\mathcal{G}(1, \mu, \sigma) - \mathcal{G}(-1, \mu, \sigma)}, \tag{15}$$

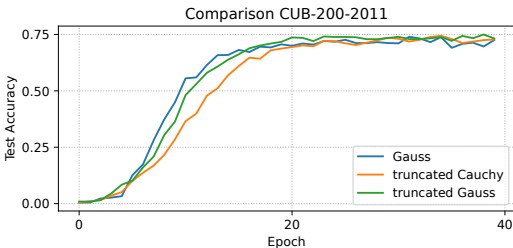

Figure 10: CUB-200-2011 Test Accuracy of HyperPgNet with different probability distributions. The difference in performance is only marginal.

with lower bound $a$ and upper bound $b$, e.g., for the cosine similarity $a = -1$ and $b = 1$. Similarly, the truncated Cauchy distribution can be applied, which is defined as

$$\mathcal{T}_{\text{Cauchy}}(x, x_0, \gamma, a, b) = \frac{1}{\gamma} \left( 1 + \left( \frac{x - x_0}{\sigma} \right)^2 \right)^{-1} \left( \arctan \left( \frac{b - x_0}{\gamma} \right) - \arctan \left( \frac{a - x_0}{\gamma} \right) \right)^{-1}. \tag{16}$$

Fig. 10 shows the test accuracy of three HyperPgNet models with Gaussian, truncated Gaussian and truncated Cauchy distribution on the CUB-200-2011 dataset. While difference in test performance and learning speed were minimal on the CUB-200-2011 dataset, further exploration is necessary, as other experiments showed that the concept-alignment on CUB-200-2011 dominates the learning process, lessening the influence of the prototypes.

### A.3 VON MISES-FISHER DISTRIBUTION

The von Mises-Fisher distribution (vMF) is the analogue of the Gaussian distribution on the surface of a hypersphere Hillen et al. (2017). The density function $f_d$ of the vMF distribution for a $D$-dimensional unit-length vector $\boldsymbol{v}$ is defined as

$$f_d(\boldsymbol{v}|\boldsymbol{\alpha}, \kappa) = C_d(\kappa) \exp \left( \kappa \boldsymbol{\alpha}^\top \boldsymbol{v} \right), \tag{17}$$

with mean vector $\boldsymbol{\alpha}$, scalar concentration parameter $\kappa$ and normalization constant $C_d(\kappa)$. The normalization constant $C_d(\kappa)$ is a complex function and difficult to compute for higher dimensions, which is why, for example, Tensorflow[1] only supports the vMF distribution for $D \leq 5$. However, the vMF distribution is a viable similarity measure when using the unnormalized density function with $C_d(\kappa) = 1$. Working with unnormalized densities highlights the relationship between the normal distribution and the vMF distribution.

Let $\hat{G}$ be the unnormalized PDF of a multivariate Gaussian with normalized mean $\boldsymbol{\alpha}$ and isotropic covariance $\boldsymbol{\sigma}^2 = \kappa^{-1} \boldsymbol{I}$, then it is proportional to the vMF distribution for normalized vectors $\boldsymbol{v}$ with

---

[1]Tensorflow API Documentation - Accessed 2024-09-20

$|\boldsymbol{v}| = 1$, as shown by

$$\hat{G}(\boldsymbol{v}|\boldsymbol{\alpha}, \kappa) = \exp\left(-\kappa\frac{(\boldsymbol{v} - \boldsymbol{\alpha})^\top(\boldsymbol{v} - \boldsymbol{\alpha})}{2}\right) \tag{18}$$

$$= \exp\left(-\kappa\frac{\boldsymbol{v}^\top\boldsymbol{v} + \boldsymbol{\alpha}^\top\boldsymbol{\alpha} - 2\boldsymbol{v}^\top\boldsymbol{\alpha}}{2}\right) \tag{19}$$

$$= \exp\left(-\kappa\frac{1 + 1 - 2\boldsymbol{v}^\top\boldsymbol{\alpha}}{2}\right) \tag{20}$$

$$= \exp\left(-\kappa\frac{2 - 2\boldsymbol{v}^\top\boldsymbol{\alpha}}{2}\right) \tag{21}$$

$$= \exp\left(-\kappa\frac{1 - \boldsymbol{v}^\top\boldsymbol{\alpha}}{1}\right) \tag{22}$$

$$= \exp\left(\kappa(\boldsymbol{v}^\top\boldsymbol{\alpha} - 1)\right) \tag{23}$$

$$= \exp(\kappa\boldsymbol{v}^\top\boldsymbol{\alpha} - \kappa) \tag{24}$$

$$= \exp(\kappa)^{-1}\exp(\kappa\boldsymbol{v}^\top\boldsymbol{\alpha}) \tag{25}$$

$$\sim \exp\left(\kappa\boldsymbol{v}^\top\boldsymbol{\alpha}\right). \tag{26}$$

Eq. 23 also shows the relationship to the HyperPg similarity with an untruncated Gaussian distribution and prototype mean activation $\mu = 1$. Fig. 11 presents a simulation of the vMF distribution on a 3D sphere. While both the vMF distribution and HyperPg activation can produce a spherical, gaussian-like activation pattern on the surface of a hypersphere, the vMF distribution cannot produce the ring pattern shown in Fig. 2. The ring pattern produced by adapting HyperPg's mean similarity $\mu$ could be approximated by a mixture of vMF distributions.

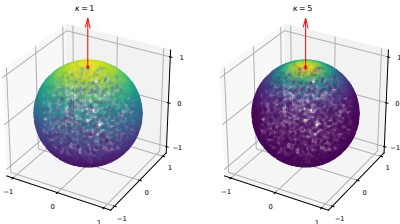

Figure 11: Changing the concentration parameter $\kappa$ is akin to changing HyperPg's std $\sigma$.

## A.4 FISHER-BINGHAM DISTRIBUTION

As the vMF distribution is the equivalent of an isotropic Gaussian distribution on the surface of a hypersphere, the Fisher-Bingham (FB) distribution is the equivalent of a Gaussian with full covariance matrix. Similar to the vMF, the normalization constant is difficult to compute for higher dimensions, but the unnormalized density function remains feasible.

For a $D$ dimensional space, the FB distribution is by a $D \times D$ matrix $\boldsymbol{A}$ of orthogonal vectors $(\boldsymbol{\alpha}_1, \boldsymbol{\alpha}_2, \ldots, \boldsymbol{\alpha}_D)$, concentration parameter $\kappa$ and ellipticity factors $[\beta]_{2:D}$ where $\sum_{j=2}^{D}\beta_j = 1$ and $0 \leq 2|\beta_j| < \kappa$. The FB unnormalzied PDF is defined as

$$b(\boldsymbol{v}|\boldsymbol{A}, \kappa, \beta) = \exp\left(\kappa\boldsymbol{\alpha}_1^\top\boldsymbol{v} + \sum_{j=2}^{D}\beta_j\left(\boldsymbol{\alpha}_j^\top\boldsymbol{v}\right)^2\right). \tag{27}$$

The FB distribution's main advantage is the elliptic form of the distribution on the surface of the hypersphere, offering higher adaptability than the other formulations (see Fig. 12. However, the parameter count and constraints are higher.

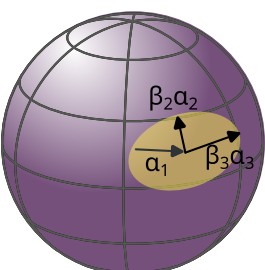

Figure 12: Illustration of the Fisher-Bingham Distribution in $D = 3$

## A.5 MIXTURE MODELS

HyperPg's probabilistic nature lends itself to a mixture formulation. Let the definition of a HyperPg Mixture Prototype be $\boldsymbol{p}^M = (\boldsymbol{\alpha}, \mu, \sigma, \pi)$ with additionally learned mixture weight $\pi$. Further, let's define the probability of a latent vector $\boldsymbol{z}$ belonging to a Gaussian HyperPg prototype $\boldsymbol{p}$ as

$$\phi(\boldsymbol{z}|\boldsymbol{p}) = s_{\text{HyperPg}}(\boldsymbol{z}|\boldsymbol{p}). \tag{28}$$

Then the probability of $\boldsymbol{z}$ belonging to class $c$ can be expressed through the mixture of all prototypes $\boldsymbol{p}_c \in \boldsymbol{P}_c$ of class $c$, i.e.,

$$\phi(\boldsymbol{z}|c) = \sum_{\boldsymbol{p}_c \in \boldsymbol{P}_c} \pi(\boldsymbol{p}_c)\phi(\boldsymbol{z}|\boldsymbol{p}_c). \tag{29}$$

First experiments with mixture of HyperPg prototypes did not show any improvement over the standard formulation. However, this might change with other datasets.

## B IMPLEMENTATION DETAILS

### B.1 DATA PREPROCESSING

In contrast to prior work (e.g. Chen et al., 2019; Rymarczyk et al., 2020; Ukai et al., 2023) the experiment used an online augmentation process, resulting in 30 training images per class and epoch. The input images were first resized to a resolution of $224 \times 224$ *without* cropping to bounding box annotations. The augmentations consisted of RandomPerspective, RandomHorizontalFlip and RandomAffine.

### B.2 HYPERPARAMETERS

The prototypical part networks ProtoPNet and HyperPgNet use a convolutional neck after the feature encoder and work on a latent feature map of size $7 \times 7 \times 256$. The models were trained with a minibatch size of 96 images using AdamW optimizer with learning rate 1e-4 and weight decay 1e-4.

### B.3 COMPUTE RESOURCES

All experiments were performed on a workstation with a single NVIDIA RTX 3090 GPU (24 GB VRAM) per model.

### B.4 MODEL IMPLEMENTATION

The models were implement with different feature encoding backbones. ConvNeXt-tiny (Liu et al., 2022) and ResNet50 (He et al., 2016) with pretrained weights on ImageNet provide CNN based backbones. As a transformer based backbone MiT-B4 (Xie et al., 2021) as provided by HuggingFace is used. MiT-B4 is pretrained on ImageNet and finetuned on ADE20k, Cityscapes and COCO-stuff.

The models are implemented as follows:

- The black box baseline model uses the pretrained feature encoder as backbone followed by the classification head: a single linear output layer and softmax activation.
- Concept Bottleneck Models (Koh et al., 2020) integrate the concept bottleneck, a linear layer with one neuron per concept, between the backbone and classification head.
- ProtoPNet (Chen et al., 2019) use a $L_2$ based prototype layer between backbone and classification head.
- ProtoPNet + HyperPg use a HyperPg based prototype module between backbone and classification head.
- HyperPgNet $-\mathcal{L}_{\mathrm{RRC}}$ uses a HyperPg prototype module between backbone and classification head. The prototypes are trained with concept alignment, but without the $\mathcal{L}_{\mathrm{RRC}}$ loss.
- HyperPgNet $+\mathcal{L}_{\mathrm{RRC}}$ uses a HyperPg prototype module between backbone and classification head. The prototypes are trained with concept alignment and including the $\mathcal{L}_{\mathrm{RRC}}$ loss.

## C  EXTENDED INTERPRETABILITY ANALYSIS

### C.1  LATENT SPACE STRUCTURE

Prototype learning is based on learning structures in the latent space. HyperPg specifically learns Gaussian distributions on the surface of a hypersphere in a high-dimensional latent space. Dimensionality reduction techniques like UMAP (McInnes et al., 2018) aim to preserve global and local structures from a high dimensional space when projecting into a low dimensional one. UMAP supports multiple distance metrics, including hyperspheric manifold distances, thus retaining some of the high dimensional structure when projecting onto a 3D sphere.

Fig. 13 illustrates UMAP projections of HyperPg concept-aligned prototypes trained on Stanford Cars. This visualization indicates that HyperPgNet is able to disentangle the different concepts in latent space, as the projection shows no overlap of the different clusters. When trained with $L_{\mathrm{RRC}}$, the prototypes are packed closer together. This could indicate, that the chosen hyper parameter of 20 prototypes per concept is higher than required as not all concept embeddings have the same diversity in the latent space.

Fig. 14 illustrates one UMAP projection of HyperPg concept-aligned prototypes trained on CUB-200-2011. In comparison to Stanford Cars, the latent space appears less structured. This could explain the higher difficulty associated with this dataset.

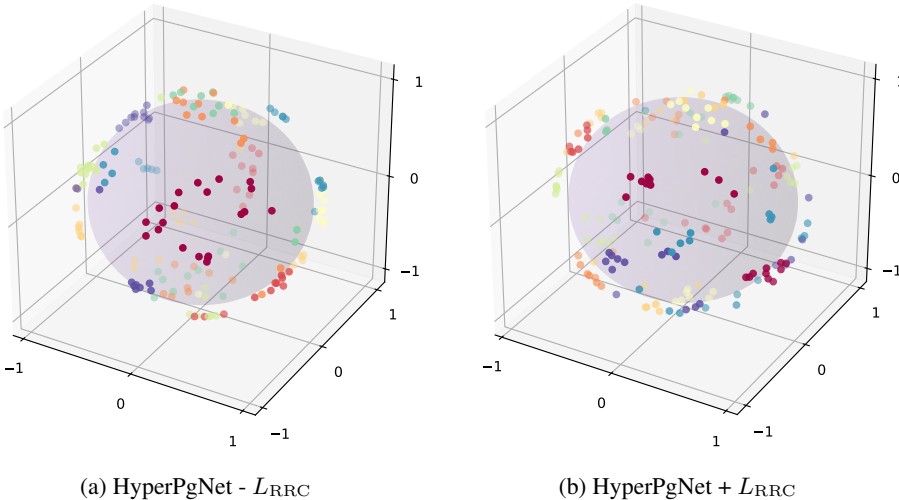

(a) HyperPgNet - $L_{\mathrm{RRC}}$                    (b) HyperPgNet + $L_{\mathrm{RRC}}$

Figure 13: UMAP projection of HyperPg concept prototypes learned on Stanford Cars. Each dot represents one prototype with the color indicating a concept.

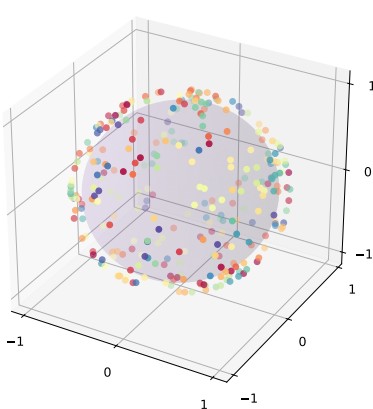

Figure 14: UMAP Projection of HyperPg concept prototypes learned on CUB-200-2011.

