# OpenReview forum: "HyperPg - Prototypical Gaussians on the Hypersphere for Interpretable Deep Learning"
_ICLR.cc/2025/Conference — Submitted to ICLR 2025_

### Official Review · Reviewer_mvjz · 2024-10-31

**Soundness:** 1
**Presentation:** 3
**Contribution:** 3
**Rating:** 1
**Confidence:** 5

**Summary:**

The paper presents a comprehensive approach to advancing prototype-based learning by introducing prototypes modeled as Gaussian mixtures and aligning them closely with underlying semantic concepts in the data. To achieve this, the authors propose a novel architecture built upon the SegFormer backbone. This architecture is enhanced by incorporating the "right for the right reasons" loss function, which emphasizes alignment between model predictions and human-understandable reasons behind those predictions, ensuring that the model learns interpretable and robust representations.

The effectiveness of the proposed approach is evaluated on well-established benchmark datasets, specifically CUB and Stanford Cars, which provide challenging tasks for fine-grained visual categorization. Moreover, to emulate realistic conditions where user interaction might guide the discovery of underlying concepts within the dataset, the authors simulate user input.

**Strengths:**

The methodology introduced in this paper is both novel and compelling. The core idea of representing prototypical parts as Gaussian mixtures stands out as an innovative approach.

The motivation behind this work is well-articulated, addressing key challenges in interpretable and concept-aligned machine learning.

Furthermore, the paper is carefully organized, with a logical progression of ideas that guides the reader through the technical details and innovations. Each section is clearly written, making complex concepts accessible and enhancing the paper's overall readability.

**Weaknesses:**

I have significant reservations regarding the evaluation methodology presented in this paper. First, the ProtoPNet model, which uses a CNN backbone, achieves notably better results than the model with the SegFormer backbone. This discrepancy is unexpected and calls for further investigation. I recommend that the authors consider using the same backbone for a more consistent comparison with ProtoPNet, as differences in backbones could introduce confounding variables that obscure the real advantages of the proposed method.

Secondly, I would like to direct the authors to two relevant publications, [1] and [2], which address crucial aspects of prototypical part learning. Publication [1] introduces a benchmark specifically designed for prototypical parts, which could help in assessing the robustness of prototypical parts learned through transformer-based architectures like SegFormer. Due to SegFormer's global attention mechanism, it may be prone to spatial misalignment.

Additionally, publication [2] presents a model that effectively disentangles visual features—such as color, shape, and texture—within prototypical parts. With the current approach, I am not convinced that the proposed model achieves this level of separation. Conducting further experiments or introducing additional benchmarks could help verify it.

Finally, I suggest that the authors compare their approach to concept bottleneck models [3]. Given that the current work intersects both concept learning and prototype alignment, a comparison with these models would provide a more comprehensive evaluation and clarify the unique contributions of the proposed method in relation to established approaches.

[1] Sacha, Mikołaj, et al. "Interpretability benchmark for evaluating spatial misalignment of prototypical parts explanations." Proceedings of the AAAI Conference on Artificial Intelligence. Vol. 38. No. 19. 2024.
[2] Pach, Mateusz, et al. "LucidPPN: Unambiguous Prototypical Parts Network for User-centric Interpretable Computer Vision." arXiv preprint arXiv:2405.14331 (2024).
[3] Koh, Pang Wei, et al. "Concept bottleneck models." International conference on machine learning. PMLR, 2020.

**Questions:**

1. Why not compare with CNN-based ProtoPNet?

2. Reference to papers mentioned in weaknesses.

3. Discussion and comparison to CBM models.

---

> ### Author Response · Authors · 2024-11-15
>
> Thanks for your valuable feedback! We appreciate that you find our idea innovative and like the overall motivation and structure of the paper.
>
> We regrettably agree, that our experiment does not sufficiently highlight the advantages of our model. We are currently running more CNN based experiments to address this issue. One point which we need to better highlight when describing our experimental setup, is the use of full images on CUB-200-2011. In contrast to other prototype learning work, we do not use crop to the bounding box for our experiments. Additionally, the experiments on Stanford Cars were not fully converged until the submission deadline. We posted a new revision with updated results, which are also more in line with prior reported results. Having said all that: We are preparing additional experiments with different backbone models.
>
> Thank you for pointing out the additional publications. They are of high interest to us and warrant further discussion. On a first, quick glance, [1] could provide an interesting test if the prototypical parts are actually aligned. Our current hope is, that the inclusion of the RRC loss could mitigate this issue during learning, and a gradient based visualization method during the explanation stage. We will look into this work more in depth.
>
> [2] does really interesting work with learning a hybrid input head to disentangle color and shape. We hope that a better qualitative evaluation will show that our model can successfully learn prototypes for concepts such as "red head" and "white head". But directly learning disentangled prototypes for "red", "white" and "head" with our current model would be a welcome surprise. We will provide a more thorough analysis of our model to strengthen our manuscript and double check we do not claim inflated contributions.
>
> We also see CBM as an important method to compare to. For CUB-200-2011, CBM's performance (75.1) is on a level with our models (75.8 & +RRC 71.9). However, it can only say if a concept is present, not where, reducing the insights it provides. Moreover, CBM struggles with multi-modal concepts, which HyperPgNet can learn by increasing the number of prototypes per concept. This is why CBM's performance on Stanford Cars is lower, because the provided concepts like "front lights" are rather broadly defined. CBM scores 79.9% compared to our methods 87.5% and +RRC 84.7%. We will highlight these results and emphasize the advantages of our model in the manuscript.
>
> Please let us know if you have any additional questions or concerns.

---

> > ### Comment · Reviewer_mvjz · 2024-11-20
> > **The work is not yet ready**
> >
> > Dear Authors,
> >
> > You did an outstanding job addressing our doubts and questions, providing additional results and clarifications along the way. However, I agree with the other reviewer (ydoP) that the work is not yet ready for publication. The volume of changes and additional experiments requested during the review process is substantial. Furthermore, as you yourself acknowledged, the Stanford Cars results had not yet converged at the time of submission.
> > I recommend taking the time to thoughtfully incorporate our feedback and consider submitting this work to a different venue, such as IJCAI, ICML, ICCV, or KDD, as their deadlines are approaching. A thorough revision will be necessary to bring the work to the level required for publication. At this stage, I do not believe it is feasible to prepare it in time for ICLR.

---

### Official Review · Reviewer_ydoP · 2024-10-31

**Soundness:** 2
**Presentation:** 2
**Contribution:** 1
**Rating:** 3
**Confidence:** 4

**Summary:**

This paper builds on top of the prototype-based learning literature and proposes to combine the usage of Gaussian and hyperspherical prototypes by learning prototypes that are defined as Gaussians on a hypersphere. The combine this with the loss from Ross et al. In order to make the prototype maps more faithful to available segmentation maps.

**Strengths:**

The paper is written in a clear manner and is easy to follow. The method is simple.

**Weaknesses:**

I’m under the impression that the authors don’t really manage to drive the message that the proposed method results in any advantage:

The experimental setup is rather limited:
1) Only two datasets with one single setting, no ablations nor tests with different number of prototypes.
2) Very few competitors, and rather outdated (2019 and 2020). Many other works are mentioned in Section 2. What is the reason not to include more of them?
3) The baselines are rather weak. ConvNext-tiny is far from a model that can obtain sota results, and a more adapted baseline would dramatically improve these results. SegFormer is a segmentation model. I’m not sure how the authors trained it for classification, but it sort of makes
4) For an inpteretability-focused paper, there are very few qualitative results. On top of that, the few that are available suggest that the prototype gradient maps are not particularly interpretable. How do there compare to competing methods?
5) Fig 5 seems to suggest that some of the baselines did not yet reach convergence.

6) Other than the lack of empirical evidence, the authors don’t provide a convincing argument of why Gaussians in a hyperphere should be better prototype representations.

**Questions:**

I think a lot more experimental evidence, along with a clearer rationale, is needed for this paper to be convincing.

---

> ### Author Response · Authors · 2024-11-15
>
> Thanks for your valuable feedback! We appreciate that you like the manuscript and the simplicity of our approach. We regret and agree that the current manuscript does not sufficiently highlight our contributions. Regarding your comments:
>
> 1. We appreciate your request for additional datasets. The chosen Datasets are in line with prior work on prototype learning such as [2], [3], [4]. However, we are now also looking at datasets such as Stanford Dogs and Oxford Flowers. We hope to provide additional results, soon. We apologize for not highlighting our ablation studies better. We present some quantitative results with different number of prototypes in Appendix B.2, although qualitative results would also be of interest.
>
> 2. We selected Concept Bottleneck models as a baseline for a model, which also incorporates concept information similar to our method. ProtoPNet is the prototype learning base model, most prototype learning models, including ours, is built upon. We will improve our manuscript to better highlight the differences between our model and other works. For example, ProtoKNN uses a KNN classifier after ProtoPNet's prototype layer instead of a linear classification layer. Or Deformable ProtoPNet learns a spatial combination of a set of smaller prototypes.
>
> 3. The choice for ConvNeXt-tiny was inspired by other recent work such as [5], [6]. Regarding Segformer: You are right, this is a segmentation model and are currently verifying our implementation. However, we believe we correctly used a classification implementation built on MiT-B4 transformer introduced in the Segformer work. As the performance of the downstream models (ProtoPNet and HyperPgNet) is comparable to other reported results, we are confident the experiments are valid. Still, we acknowledge that this causes unnecessary confusion and distracts from our contribution. We are currently running more CNN based experiments, but hope to also provide additional results with a different, transformer based backbone.
>
> 4. This is a shared sentiment among all reviewers, which we regretfully agree with. We are working on a more extensive evaluation and hope to provide results soon. A comparison with other methods would be further strenghten our work.
>
> 5. We apologize for not finishing the experiments you mentioned in time for the first manuscript submission. We already updated a revision with converged results. As you will notice, with enough training time, the other methods also reach comparable results. However, we can already see that HyperPgNet und HyperPg prototypes require fewer training epochs to achieve this level of performance. Additionally, we hope to show the increase in interpretability by providing more qualitative results.
>
> 6. We will update the manuscript to better motivate our approach and provide more meaningful results.
>
> We hope that we could answer your questions and we will keep you updated with new results as they come in.
>
> [2] Chen et al 2019 - ProtoPNet https://proceedings.neurips.cc/paper/2019/hash/adf7ee2dcf142b0e11888e72b43fcb75-Abstract.html
>
> [3] Donelly et al 2022 - Deformable ProtoPNet https://openaccess.thecvf.com/content/CVPR2022/papers/Donnelly_Deformable_ProtoPNet_An_Interpretable_Image_Classifier_Using_Deformable_Prototypes_CVPR_2022_paper.pdf
>
> [4] Ukai et al 2023 - ProtoKNN https://openreview.net/forum?id=lh-HRYxuoRr
>
> [5] Wang et al. 2024 - MCPNet https://eddie221.github.io/MCPNet/
>
> [6] Nauta et al 2023 - PIPNet https://openaccess.thecvf.com/content_ICCV_2019/papers/Hwang_SegSort_Segmentation_by_Discriminative_Sorting_of_Segments_ICCV_2019_paper.pdf

---

> > ### Comment · Reviewer_ydoP · 2024-11-16
> >
> > I would like to start by commending the authors for their transparency and diplomacy with which they have answered to all the reviewers, which is highly appreciated.
> >
> > I also thank them for the clarifications for their choices in Q2 and Q3.
> >
> > I sincerely hope the comments from this revision round helps the authors improve their ongoing work, which is, in my opinion, not yet ready for publication at ICLR.

---

### Official Review · Reviewer_dYKD · 2024-11-03

**Soundness:** 3
**Presentation:** 3
**Contribution:** 2
**Rating:** 5
**Confidence:** 4

**Summary:**

In this paper, the authors propose HyperPg, , a new prototype representation leveraging Gaussian distributions on a hypersphere in latent space, with learnable mean and variance for adapting to the spread of clusters in the latent space and outputs likelihood scores. Additionally, the authors propose HyperPgNet, which leverages HyperPg to learn prototypes aligned with human concepts from pixel-level annotations. Experiments on CUB-200-2011 and Stanford Cars datasets demonstrate that HyperPgNet outperforms other
prototype learning architectures while using fewer parameters and training steps.

**Strengths:**

1. The idea of learning explainable and interpretable prototypes is interesting and well-motivated.
2. The paper is generally well-written and well organized.

**Weaknesses:**

1. The novelty of the paper is somewhat limited. The idea of using Gaussian distribution for prototype learning has been extensively studied before in computer vision. Please see [1] for an example. The authors need to clarify the difference of the proposed method with these existing works.

2. The experiments are limited. Only two datasets are included. Also, the authors leverage the Grounding DINO model and SAM for auto-labelling, it is not clear if the labels obtained in this manner are actually accurate enough to learn the prototype. Therefore, the authors need to clarify if the conclusions are supported by the results.






[1] SegSort: Segmentation by Discriminative Sorting of Segments

**Questions:**

1. What are the main benefits of the proposed method? In Table 1, it can be observed that the proposed method is better than black box methods on Stanford cars, but actually comparable on CUB-200-2011. Also, qualitative results are rather limited. Therefore, it is not clear why the proposed method can increase interpretability of the model.

2. How are the hyperparameters selected in the proposed loss?

3. Can the authors provide more analysis on the  interpretability of the model? Seems like the propose method can locate the meaningful regions from the images, what is the difference of this compared with the attention mechanism in transformer architecture which can also achieve similar effects.

4. In Table 1, why the segformer baseline achieve much worse results compared with other methods?

---

> ### Author Response · Authors · 2024-11-15
>
> Thank you for the valuable insights and feedback. We appreciate that you find our paper well written and our method interesting. One common concern amongst the reviewers is the need for more extensive quantitative and qualitative experiments, with which we wholeheartedly agree. We are taking the feedback to start additional experiments, for which we will provide the results as soon as they come in.
>
> Thank you for pointing out the reference. We will add it to the related work section. For a somewhat short comparison, we focus on the proposed von Mises Fisher Distribution (vMF) and our HyperPg activation:
>
> 1. We added a Section to the Appendix (A.3) which shows the equivalency of the unnormalized vMF distribution, as used by [1], with the HyperPg activation with fixed $\mu=1$. Please note, that the vMF mean **vector** $\mu$ corresponds to the HyperPg anchor vector $\alpha$, not HyperPg's Gaussian mean **skalar** $\mu$. Because HyperPg can optimize and learn the Gaussian mean, as it is not fixed at 1, which leads to the complex activation patters as depicted in Fig 1.
>
> 2. From an implementation standpoint, the formulation of HyperPg as a two step approach with some anchor similarity and learned probability function enables future extensions. By using a different manifold and similarity metric, different properties could be enforced. Or a different, multi-modal probability distribution could be used instead of the Gaussian distribution.
>
> We appreciate your request for additional datasets. The chosen Datasets are in line with prior work on prototype learning such as [2], [3], [4]. However, we are now also looking at datasets such as Stanford Dogs and Oxford Flowers. We hope to provide additional results, soon. We hope to show with more qualitative results that the quality of the labels are sufficient for HyperPgNet to learn prototypes aligned with the human defined concepts.
>
> Regarding your questions:
> 1. As a different reviewer noted, the results on Stanford Cars were unfortunately not converged in time. With enough training time, the other methods also reach comparable results. However, we can already see that HyperPgNet und HyperPg prototypes require fewer training epochs to achieve this level of performance. Additionally, we hope to show the increase in interpretability by providing more qualitative results. A model with black-box model level of performance and inherent interpretability will hopefully enable downstream tasks with increased requirements for model trust in safety critical applications
> 2. We apologize for our oversight of not including the hyperparameters in the first manuscript. For ProtoPNet, we follow the hyperparameters proposed by the authors, with 0.8 and 0.08 for cluster and separation loss. For the newly introduced HyperPg prototypes, all loss terms are weighted equally at 1. We have added this information in the revision.
> 3. Thank you for highlighting this issue. Currently, our qualitative results do not sufficiently convey the capabilities of our model. We are preparing and updated result section and aim at explanations in the manner of: "This input is classified as A, because this image region is activated by the prototype for concept B. The concept B can also be found in these regions from known training images...". We agree that the currently presented results make our contribution look like a common saliency map or attention map. We hope, updated results will better demonstrate the capabilities of our method.
> 4. Despite running multiple hyperparameter sweeps, we were unable to achieve better results. However, we feel the results are still valid, as the downstream models still performed well. Although, following feedback from all the comments, we hope to run additional experiments with other backbone models.
>
> We hope that we could answer your questions and we will keep you updated with new results as they come in.
>
> Edit: Forgot to add the references.
> [2] Chen et al 2019 - This looks like that (ProtoPNet) https://proceedings.neurips.cc/paper/2019/hash/adf7ee2dcf142b0e11888e72b43fcb75-Abstract.html
> [3] Donelly et al 2022 - Deformable ProtoPNet https://openaccess.thecvf.com/content/CVPR2022/papers/Donnelly_Deformable_ProtoPNet_An_Interpretable_Image_Classifier_Using_Deformable_Prototypes_CVPR_2022_paper.pdf
> [4] Ukai et al 2023 - ProtoKNN - https://openreview.net/forum?id=lh-HRYxuoRr

---

> ### Comment · Reviewer_dYKD · 2024-12-02
>
> Thanks for providing the rebuttal. However, the current form of the paper lacks enough evidence to show the benefits of the proposed method, therefore I will maintain my current rating.

---

### Official Review · Reviewer_DKS3 · 2024-11-04

**Soundness:** 2
**Presentation:** 1
**Contribution:** 3
**Rating:** 3
**Confidence:** 4

**Summary:**

This paper argues that conventional prototype methods often capture latent prototypes without human control, resulting in unstable and less transparent prototype generation. To address these issues, the paper introduces Gaussian distribution into the prototypes and proposes a novel optimization mechanism for refining them, enabling more reasonable and interpretable results.

**Strengths:**

1. The idea presented in this paper is both motivated and novel, with the proposed method aligning well with the underlying motivation.
2. The quantitative and qualitative results demonstrate the effectiveness of the proposed method to some extent.

**Weaknesses:**

1. The organization of this paper requires significant improvement, as the content's logic is disjointed. For instance, in the methodology section, the paper introduces the training loss first, then unexpectedly shifts to the model structure before resuming the discussion on the loss and proposed improvements. Additionally, when describing traditional prototype methods, the paper does not seamlessly explain the necessity of the Gaussian distribution, making the argument unconvincing.
2. The paper should include more quantitative and qualitative results in the experimental section. Currently, it dedicates too many pages to the method description while lacking sufficient experimental outcomes. It is recommended to present additional qualitative results in the Appendix.
3. This paper should provide a more detailed discussion on the challenges of training the proposed prototype generation method. Prototype learning is known as difficult to train, yet the paper only briefly mentions the complexity of the proposed method without offering in-depth analysis.

**Questions:**

See the weakness part.

---

> ### Author Response · Authors · 2024-11-15
>
> Thank you for your valuable feedback. One common concern amongst the reviewers is the need for more extensive quantitative and qualitative experiments, with which we wholeheartedly agree. We are taking the feedback to start additional experiments, for which we will provide the results as soon as they come in.
>
> In the meantime we will improve the clarity of the manuscript and address the points you mentioned. As a first change, we reordered the sections describing the losses to improve the logical structure, and will continue to improve the motivation for our approach in contrast to traditional prototype methods.
>
> Regarding your third point: We added some more details regarding the choice of hyper-parameters (equally weighted for HyperPg prototypes), but do not feel confident we answered your question. Could you please clarify, what kind of complexity you mean?

---

> > ### Comment · Reviewer_DKS3 · 2024-12-02
> >
> > As I mentioned before, this work lacks sufficient experimental results, so it is not finished yet. I wish a better future exploration.

---

### Author Response · Authors · 2024-11-15

We thank all Reviewers for their valuable feedback and insights. The main concern regarding our paper seems to be the lack of evaluation and experiments to support our motivation, with which we agree. We will conduct extensive experiments over the next few days to showcase the advantages of our approach.

We will address your questions in detailed comments under each review. Additionally, we have prepared an initial update to our manuscript incorporating the first set of changes proposed by the reviewers. One significant change is the way we describe our experiments, which we hope improves the clarity and highlights our contributions more effectively.

Section 6, Experiments, now lists the compared models as:
- Segformer MiT-B4
- ConvNeXt-tiny
- Concept Bottleneck Models (CBM)
- ProtoPNet with $L_2$ prototypes
- ProtoPNet with HyperPg prototypes
- HyperPgNet with concept aligned prototypes
- HyperPgNet with concept aligned prototypes and $L_{RRC}$ optimization

We are currently working on extensive experiments using CNN backbone models and a better qualitative analysis to highlight our model's interpretability. We will provide the additional results as soon as they are ready.

Again, we appreciate everyone's input and look forward to strengthening our manuscript further.

---

### Author Response · Authors · 2024-11-27
**Final Revision**

Dear Reviewers,

we thank you again for your invaluable feedback and kind responses. Over the last few days, we have been preparing a revision of our manuscript, striving to incorporate as many of your comments as possible. We believe your questions and comments have significantly helped us in highlighting our contribution. In summary, our changes are as follows:

**Major Revision of Result Section 6**: Based on your feedback, we rewrote our result section to include more quantitative and qualitative results. We hope, this revision helps to confirm and highlight our contribution.

To free up the required space for the expanded result section, we also made some minor changes:
- Reworded the Related Work section to make it more concise and better differentiate our method from existing ones.
- Clarified the Conclusion section.
- Elaborated on the implementation details, but moved them to the appendix, similar to other works. We added a reproducibility statement to make it easier for other researchers to find the relevant sections for re-implementation.
- Made several improvements in the appendix.


As all reviewers identified the results section in the first manuscript as a weak point, we kindly request you to read **Section 6: Experiments** and consider if a change in score is justified. We are looking forward to answering any new questions and concerns throughout the extended discussion phase.

Regardless of the outcome of this review phase, we feel encouraged by the reception of our proposed method and will continue working in this direction. We sincerely express our thanks and wish everyone good luck with their own submissions.

---

### Meta-Review · Area_Chair_uqGw · 2024-12-14

**Metareview:**

This paper investigates the use of Gaussian prototypes for interpretable image classification, highlighting their potential to enhance model transparency. The proposed approach demonstrates incremental improvements when integrated with the RRC loss, offering a novel perspective on model training dynamics. However, the experimental results presented are relatively limited in scope, leaving room for further evaluation. To solidify the contributions of this work, a more comprehensive set of comparisons with existing explainable AI (XAI) methodologies is essential, providing a clearer understanding of its relative effectiveness and practical applicability.

**Additional Comments On Reviewer Discussion:**

After revision, the reviewers are not satisfied with the experimental results.

---

### Decision · Program_Chairs · 2025-01-22

Reject